# Unchosen Experts Can Contribute Too: Unleashing MoE Models' Power by Self-Contrast

**Chufan Shi**[1*]  **Cheng Yang**[1*]  **Xinyu Zhu**[2*]  **Jiahao Wang**[3*]
**Taiqiang Wu**[3]  **Siheng Li**[1]  **Deng Cai**[4]  **Yujiu Yang**[1†]  **Yu Meng**[2†]
[1]Tsinghua University  [2]University of Virginia
[3]The University of Hong Kong  [4]Tencent AI Lab
scf22@mails.tsinghua.edu.cn
yang.yujiu@sz.tsinghua.edu.cn    yumeng5@virginia.edu

## Abstract

Mixture-of-Experts (MoE) has emerged as a prominent architecture for scaling model size while maintaining computational efficiency. In MoE, each token in the input sequence activates a different subset of experts determined by a routing mechanism. However, the unchosen experts in MoE models do not contribute to the output, potentially leading to underutilization of the model's capacity. In this work, we first conduct exploratory studies to demonstrate that increasing the number of activated experts does not necessarily improve and can even degrade the output quality. Then, we show that output distributions from an MoE model using different routing strategies substantially differ, indicating that different experts do not always act synergistically. Motivated by these findings, we propose **S**elf-**C**ontrast **M**ixture-**o**f-**E**xperts (SCMoE), a training-free strategy that utilizes unchosen experts in a self-contrast manner during inference. In SCMoE, the next-token probabilities are determined by contrasting the outputs from strong and weak activation using the same MoE model. Our method is conceptually simple and computationally lightweight, as it incurs minimal latency compared to greedy decoding. Experiments on several benchmarks (GSM8K, StrategyQA, MBPP and HumanEval) demonstrate that SCMoE can consistently enhance Mixtral 8x7B's reasoning capability across various domains. For example, it improves the accuracy on GSM8K from 61.79 to 66.94. Moreover, combining SCMoE with self-consistency yields additional gains, increasing major@20 accuracy from 75.59 to 78.31.

## 1 Introduction

Scaling up model parameters, dataset size and training time has been considered the most direct and effective approach to improving foundation models' performance [1–3]. However, scaling dense models substantially increases computational costs, which poses a significant practical challenge. Mixture-of-Experts (MoE) [4–9] has emerged as a compelling solution for optimizing the balance between model capacity and computation overhead in the era of large foundation models.

MoE models achieve the goal by sparsely activating only a portion of the parameters for each specific input. Specifically, in MoE models, parameters are grouped into a bunch of experts, MoE models only activate some of them for processing a given input. This selective activation is achieved through a routing mechanism that dispatches each input token to a fixed number of experts (e.g, top-$k$ routing [6, 8, 10, 11]). Therefore, compared to their dense counterparts, MoE models enjoy more efficient training with significantly reduced computational costs [5–9, 11]. At the inference stage, they typically adhere to the same routing strategy as the training stage, activating only a small fraction

---

*Equal Contribution. Source code is available at `https://github.com/DavidFanzz/SCMoE.git`
†Corresponding authors.

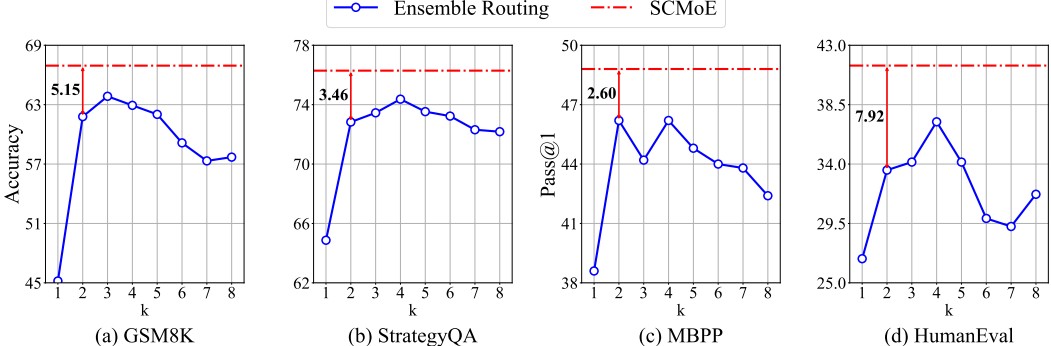

Figure 1: Performance comparison between increasing the value of top-$k$ (*i.e.*, ensemble routing) and SCMoE. SCMoE surpasses the performance of ensemble routing across various benchmarks.

of experts. Basically, for each input token, most of the well-trained experts do not contribute to the output prediction. As a result, the potential of utilizing more experts during the inference stage to enhance performance remains underexplored.

In this paper, we investigate the impact of unchosen experts[3] on the performance of MoE models and explore their suitable usage. A direct hypothesis is that incorporating more experts improves MoE models and helps solve more difficult problems [12–14]. However, in our exploratory experiment on Mixtral 8x7B [6], we find simply raising the number of activated experts (blue lines in Figure 1) does not lead to stable improvements and may even hurt performance on different tasks. This indicates that unchosen experts may contribute little or even negatively to the final performance, which is contrary to the common perception of unchosen experts as candidates of positive power.

Inspired by the finding, we further dive deep into the difference between the output probability distributions of MoE models applying different routing strategies. As shown in Figure 3, we calculate the Kullback-Leibler Divergence (KLD) between the token distributions obtained from the default top-2 routing and rank-$k$ routing, and find apparent discrepancy. The discrepancy is particularly evident in the parts that require rigorous reasoning. This suggests that different experts do not always act synergistically; instead, they may exhibit conflicting behaviors.

Therefore, we introduce **S**elf-**C**ontrast **M**ixture-**o**f-**E**xperts (SCMoE), which can convert the negative effects brought by unchosen experts into positive ones through contrasting the output logits obtained using different routing strategies. Specifically, the probability of next token is based on the logits difference between strong and weak activation of the MoE models. For "strong activation" and "weak activation", we use the top-2 routing strategy (Figure 2 (a)) and the rank-$k$ routing strategy (Figure 2 (b)) respectively. Thus, SCMoE enables unchosen experts to contribute to the prediction. An overview of how SCMoE works is presented in Figure 2 (c).

Experimental results on various benchmarks across different domains demonstrate that SCMoE significantly enhances Mixtral 8x7B's reasoning capability (Section 3). Specifically, compared to greedy decoding, the accuracy increases from 61.79 to 66.94 (+5.15) on GSM8K, 72.83 to 76.29 (+3.46) on StrategyQA, and the pass@1 accuracy increases from 46.20 to 48.80 (+2.60) on MBPP and 33.54 to 41.46 (+7.92) on HumanEval. Further analysis shows that SCMoE can even surpass the result of using self-consistency with major@5 (66.87) on GSM8K. What's more, combining SCMoE with self-consistency can further boost the model's performance, improving major@20 accuracy from 75.59 to 78.31 (+2.72) on GSM8K. Regarding inference efficiency, it turns out that SCMoE incurs only a minor (x1.30) delay compared to greedy decoding, which is competitive among several strong decoding baselines. To sum up, empirical results and comprehensive analyses demonstrate that SCMoE is a both effective and efficient approach to unleashing MoE models' power.

## 2   Method

In this section, we first provide a preliminary introduction of MoE models. Then, we present an analysis based on next-token distribution KLD to reveal the divergence between different routing

---

[3] Unchosen experts refer to the experts not selected by default routing (*e.g.*, top-2 routing in Mixtral 8x7B).

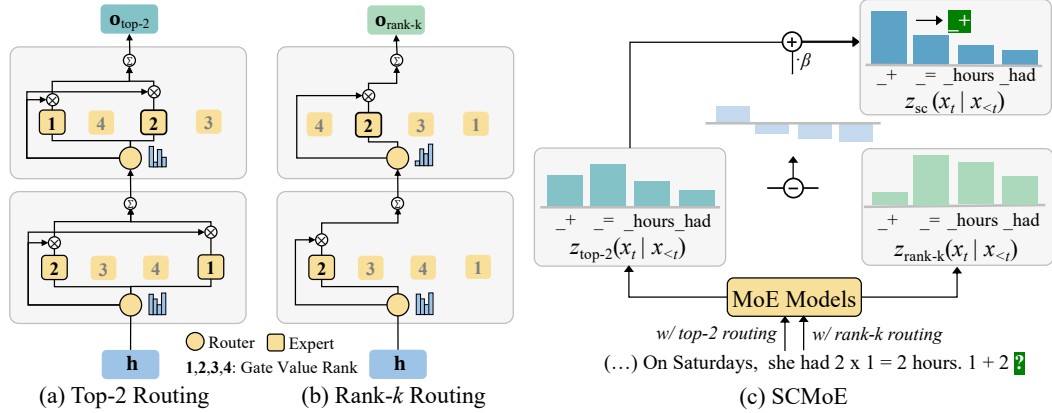

Figure 2: (a & b) Given an input $\mathbf{h}$, (a) and (b) demonstrate the workflows of top-2 routing and rank-$k$ routing (*e.g.*, k=2). We use two MoE layers as a simple schematic, omitting other layers in MoE models. Note that, in the second MoE layer, rank-$k$ routing activates the unchosen expert in top-2 routing; (c) An illustrative example of how SCMoE works, which contrasts $z_{\text{top-2}}(x_t|x_{<t})$ with $z_{\text{rank-k}}(x_t|x_{<t})$. The complete question and answer for this example are shown in Figure 3.

strategies in MoE models. This analysis motivates the introduction of SCMoE, a self-contrast method to leverage the contrastive information existing between different routing strategies in MoE models.

## 2.1 Preliminary

In Transformer-based MoE models, the conventional Feed-Forward Network (FFN) is substitued with the MoE layer [15]. Typically, each MoE layer consists of a router $R$ and a set of experts $\{E_i\}_{i=1}^{N}$. For a given input sequence $x_{<t} = (x_1, x_2, ..., x_{t-1})$, the router allocates each token in $x_{<t}$ to a specific subset of experts, which are subsequently activated to process the tokens. Specifically, given each token's hidden state $\mathbf{h}$, the router first calculates an initial gate value vector $\mathbf{w}$ across the $N$ experts as follows:

$$\mathbf{w} = \text{Softmax}(\mathbf{W}_r \mathbf{h}) \qquad (1)$$

where $\mathbf{W}_r$ denotes the weight matrix of the router. Each element $\mathbf{w}_i$ in $\mathbf{w}$ represents the probability of activating the $i$-th expert.

After that, the router applies a routing strategy (*e.g.*, top-2 or rank-$k$ routing in Section 2.2) to determine the subset of experts to be activated. Then the $\mathbf{w}_i$ of the unchoosen expert is set to 0 and $\mathbf{w}$ is renormalized to $\hat{\mathbf{w}}$ accordingly. Subsequently, the output $\mathbf{o}$ of the MoE layer is computed as the weighted sum of outputs from the activated experts:

$$\mathbf{o} = \sum_{i \in \{j | \hat{\mathbf{w}}_j \neq 0\}} \hat{\mathbf{w}}_i \cdot E_i(\mathbf{h}) \qquad (2)$$

Once the input sequence $x_{<t}$ has undergone a complete forward pass through the MoE model, the next-token distribution $p(x_t|x_{<t})$ is computed based on the output of the final layer. A decoding algorithm is then applied to predict $x_t$ from the vocabulary $\mathcal{V}$ based on $p(x_t|x_{<t})$.

## 2.2 Divergence Between Different Routing Strategies: An Exploratory Analysis

As depicted in Figure 1, unchosen experts may contribute little or even negatively to the final performance. Based on this finding, we are inspired to study the difference of output probabilities using different routing strategies. Specifically, we conduct an analysis on Mixtral 8x7B [16], with two different routing strategies, *i.e.*, top-2 routing and rank-$k$ routing, which are detailed as follows.

**Top-2 Routing**. Top-2 routing (Figure 2 (a)) [4] is the default routing strategy of Mixtral 8x7B, which activates the two experts with the highest values in $\mathbf{w}$. In this setting, the renormalized gate value for the $i$-th expert, $\hat{\mathbf{w}}_i$, is defined as follows:

$$\hat{\mathbf{w}}_i = \begin{cases} \frac{\mathbf{w}_i}{\sum_{j \in \text{top}(\mathbf{w}, 2)} \mathbf{w}_j}, & i \in \text{top}(\mathbf{w}, 2) \\ 0, & i \notin \text{top}(\mathbf{w}, 2) \end{cases} \qquad (3)$$

Question: Hallie had dance practice for 1 hour on Tuesdays and 2 hours on Thursdays. On Saturdays, she had dance practice that lasted twice as long as Tuesday's night class. How many hours a week did she have dance practice?
Answer: On Tuesdays, she had 1 hour. On Thursdays, she had 2 hours. On Saturdays, she had 2 x 1 = 2 hours. 1 + 2 + 2 = 5. The answer is 5.

| k | _On | _T | ues | days | , | _she | _had | _ | 1 | _hour | . | _On | _Th | urs | days | , | _she | _had | _ |
|---|------|------|------|------|------|------|------|------|------|------|------|------|------|------|------|------|------|------|------|
| 1 | 0.58 | 0.13 | 0.00 | 0.00 | 0.07 | 0.08 | 0.30 | 0.22 | 0.01 | 0.03 | 0.26 | 0.15 | 0.41 | 0.01 | 0.00 | 0.03 | 0.03 | 0.03 | 0.02 |
| 2 | 4.47 | 9.08 | 0.80 | 7.49 | 0.83 | 2.37 | 1.82 | 3.68 | 10.14 | 6.22 | 4.67 | 5.74 | 0.98 | 0.60 | 1.61 | 0.32 | 0.57 | 0.56 | 4.24 |
| 3 | 13.28 | 21.01 | 32.38 | 18.95 | 2.28 | 4.01 | 10.01 | 3.77 | 14.40 | 24.97 | 7.13 | 11.05 | 20.52 | 14.14 | 7.24 | 0.72 | 1.14 | 0.72 | 9.29 |
| 4 | 10.25 | 15.63 | 42.83 | 13.24 | 2.13 | 8.69 | 2.13 | 6.83 | 19.38 | 12.78 | 6.99 | 10.80 | 12.69 | 42.44 | 33.53 | 1.25 | 2.22 | 1.66 | 4.71 |
| 5 | 19.47 | 20.13 | 41.64 | 25.90 | 4.44 | 11.03 | 4.57 | 5.81 | 8.91 | 17.86 | 12.81 | 14.58 | 22.04 | 41.81 | 25.01 | 1.59 | 7.06 | 3.93 | 10.35 |
| 6 | 13.13 | 23.31 | 37.83 | 29.38 | 10.13 | 10.82 | 13.83 | 13.06 | 14.18 | 28.37 | 10.11 | 15.50 | 18.38 | 43.11 | 43.28 | 2.78 | 21.61 | 8.86 | 10.05 |
| 7 | 14.01 | 25.12 | 44.22 | 38.95 | 11.00 | 12.55 | 25.68 | 13.43 | 9.77 | 24.24 | 22.05 | 16.19 | 30.56 | 40.17 | 23.84 | 7.92 | 13.07 | 4.34 | 12.55 |
| 8 | 17.44 | 28.31 | 44.39 | 39.08 | 22.93 | 25.15 | 22.92 | 12.74 | 27.01 | 27.95 | 25.31 | 25.77 | 28.56 | 47.76 | 19.16 | 19.52 | 24.71 | 9.18 | 18.52 |

Token

| k | 2 | _hours | . | _On | _S | atur | days | , | _she | _had | _ | 2 | _x | _ | 1 | _= | _ | 2 | _hours |
|---|------|------|------|------|------|------|------|------|------|------|------|------|------|------|------|------|------|------|------|
| 1 | 0.00 | 0.05 | 0.05 | 0.24 | 0.07 | 0.00 | 0.00 | 0.02 | 0.08 | 0.11 | 0.08 | 0.13 | 0.36 | 0.03 | 0.00 | 0.09 | 0.00 | 0.00 | 0.03 |
| 2 | 0.14 | 2.21 | 6.59 | 6.67 | 1.65 | 0.14 | 3.03 | 0.70 | 0.92 | 0.51 | 6.65 | 3.77 | 3.99 | 1.12 | 3.42 | 5.25 | 1.25 | 17.88 | 4.21 |
| 3 | 5.26 | 17.42 | 8.47 | 5.95 | 17.19 | 27.46 | 29.35 | 0.33 | 0.66 | 0.79 | 4.28 | 8.15 | 9.72 | 2.69 | 14.61 | 11.23 | 1.94 | 22.71 | 11.89 |
| 4 | 2.56 | 3.25 | 15.87 | 7.81 | 19.88 | 57.47 | 20.38 | 0.77 | 1.29 | 3.45 | 3.46 | 3.68 | 8.30 | 7.03 | 15.62 | 18.09 | 4.16 | 4.55 | 25.02 |
| 5 | 10.34 | 5.22 | 13.25 | 8.61 | 17.92 | 35.02 | 9.39 | 1.61 | 1.54 | 14.45 | 7.57 | 3.53 | 12.02 | 14.45 | 14.18 | 25.35 | 7.75 | 18.49 | 19.00 |
| 6 | 12.56 | 9.27 | 11.91 | 26.61 | 17.27 | 44.09 | 35.89 | 2.06 | 26.89 | 12.76 | 11.40 | 8.59 | 16.22 | 26.15 | 25.50 | 27.20 | 35.83 | 20.21 | 24.88 |
| 7 | 12.64 | 25.72 | 16.99 | 25.64 | 23.05 | 51.35 | 14.18 | 10.16 | 12.97 | 15.87 | 7.70 | 6.52 | 16.72 | 34.41 | 29.18 | 19.97 | 9.31 | 8.85 | 18.95 |
| 8 | 8.33 | 28.44 | 22.56 | 13.88 | 26.40 | 49.99 | 39.33 | 10.76 | 18.28 | 13.59 | 8.75 | 2.11 | 20.45 | 41.83 | 23.79 | 26.50 | 10.79 | 18.05 | 26.50 |

Token

| k | . | _ | 1 | _+ | _ | 2 | _+ | _ | 2 | _= | _ | 5 | . | _The | _answer | _is | _ | 5 | . |
|---|------|------|------|------|------|------|------|------|------|------|------|------|------|------|------|------|------|------|------|
| 1 | 0.11 | 0.23 | 0.03 | 0.01 | 0.00 | 0.00 | 0.00 | 0.00 | 0.00 | 0.03 | 0.00 | 0.00 | 0.11 | 0.06 | 0.01 | 0.00 | 0.00 | 0.00 | 0.09 |
| 2 | 7.59 | 3.59 | 10.36 | 8.58 | 0.25 | 0.11 | 18.14 | 1.15 | 8.48 | 0.43 | 15.52 | 1.13 | 2.80 | 0.20 | 0.66 | 0.95 | 1.02 | 2.44 | |
| 3 | 3.43 | 6.78 | 12.83 | 13.76 | 1.15 | 5.11 | 19.74 | 2.72 | 13.61 | 6.19 | 2.03 | 14.49 | 5.94 | 5.31 | 0.61 | 0.88 | 3.76 | 8.08 | 2.67 |
| 4 | 16.16 | 7.98 | 15.31 | 18.47 | 1.30 | 1.58 | 19.35 | 3.72 | 8.46 | 8.07 | 4.83 | 17.82 | 14.10 | 5.57 | 4.53 | 2.11 | 10.26 | 21.14 | 3.86 |
| 5 | 5.80 | 7.85 | 9.87 | 23.86 | 4.19 | 9.81 | 23.53 | 15.80 | 13.13 | 12.40 | 3.85 | 15.49 | 18.82 | 11.91 | 8.14 | 0.65 | 7.20 | 23.01 | 10.11 |
| 6 | 8.10 | 13.57 | 15.68 | 24.17 | 6.80 | 32.52 | 32.04 | 13.25 | 39.56 | 16.31 | 4.77 | 21.47 | 20.94 | 13.39 | 10.49 | 9.27 | 9.84 | 22.76 | 7.47 |
| 7 | 14.89 | 22.39 | 13.63 | 27.97 | 25.11 | 2.68 | 28.18 | 14.38 | 11.06 | 17.52 | 10.43 | 17.54 | 17.56 | 16.11 | 14.00 | 9.06 | 18.09 | 21.76 | 19.00 |
| 8 | 12.70 | 19.22 | 20.47 | 30.22 | 11.78 | 33.42 | 28.72 | 9.64 | 20.80 | 27.48 | 11.88 | 13.19 | 21.87 | 24.01 | 35.70 | 2.29 | 13.41 | 38.88 | 22.66 |

Token

Figure 3: Heatmap of Kullback-Leibler Divergence between the output distribution of top-2 routing strategy ( $p_{\text{top-2}}(x_t|x_{<t})$ ) and different rank-$k$ routing strategies ( $p_{\text{rank-k}}(x_t|x_{<t})$ ). The $k$ in rank-$k$ routing ranges from 1 to 8. The values in the heatmap are scaled by $10^5$. This example is taken from the GSM8K dataset. An additional quantitative study of the KLD is provided in Appendix A.

where top($\mathbf{w}, 2$) returns the indices of the largest 2 elements in $\mathbf{w}$.

**Rank-$k$ Routing**. The rank-$k$ routing strategy (Figure 2 (b)) only activates one expert, whose initial gate value is ranked at $k$ in $\mathbf{w}$. The renormalized gate value $\hat{\mathbf{w}}_i$ is defined as follows:

$$\hat{\mathbf{w}}_i = \begin{cases} 1, & i = \text{rank}(\mathbf{w}, k) \\ 0, & i \neq \text{rank}(\mathbf{w}, k) \end{cases} \quad (4)$$

where rank($\mathbf{w}, k$) returns the index of the $k$-th largest element in $\mathbf{w}$. For Mixtral 8x7B, $k$ is enumerated from 1 to 8. We employ rank-$k$ routing as a representative strategy to utilize unchosen experts of top-2 routing (Additional statistics on the utilization ratio of unchosen experts are provided in Appendix B).

Given an input sequence $x_{<t}$, we denote the next-token distributions using top-2 routing and rank-$k$ routing as $p_{\text{top-2}}(x_t|x_{<t})$ and $p_{\text{rank-k}}(x_t|x_{<t})$, respectively. Then, we compute the Kullback-Leibler Divergence (KLD) between $p_{\text{top-2}}(x_t|x_{<t})$ and $p_{\text{rank-k}}(x_t|x_{<t})$ on GSM8K dataset. A qualitative illustration is presented in Figure 3 and a more detailed quantitative study is included in Appendix A. As shown in Figure 3, the KLD between $p_{\text{top-2}}(x_t|x_{<t})$ and $p_{\text{rank-1}}(x_t|x_{<t})$ is relatively minor, suggesting a high similarity in their next-token prediction capabilities. However, for $k$ values ranging from 2 to 8, we identify three key findings:

**Finding 1:** $p_{\text{rank-k}}(x_t|x_{<t})$ with different $k$ values exhibits discernible KLD with $p_{\text{top-2}}(x_t|x_{<t})$. As $k$ increases from 2 to 8, the KLD increases accordingly. This finding indicates the overall next-token prediction capability gap between top-2 and rank-$k$ routing.

**Finding 2:** Apparent KLD is observed when generating reasoning sequences, such as mathematical expressions (*e.g.*, "1+2+2=5") and the initiation of reasoning steps (*e.g.*, "On Thursday"). This suggests notable differences between top-2 and rank-$k$ routing in generating tokens for reasoning.

**Finding 3:** For the generation of function words (*e.g.*, "is") and punctuations (*e.g.*, ","), the KLD between $p_{\text{top-2}}(x_t|x_{<t})$ and $p_{\text{rank-k}}(x_t|x_{<t})$ is relatively smaller than that for generating critical reasoning sequences. This indicates that such predictions pose fewer challenges for rank-$k$ routing.

To sum up, it is observed that, in scenarios demanding reasoning capability for next-token prediction, MoE models with top-2 and rank-$k$ routing strategies demonstrate distinct generation behaviors. This phenomenon suggests that different experts do not always act synergistically, and could in fact exhibit conflicting behaviors. To harness such information introduced by more experts, a feasible approach is to apply contrastive methods [17, 18] to transform the observed negative impacts into positive ones.

*Therefore, we propose to leverage the contrastive information existing between different routing strategies of the MoE model (e.g., top-2 routing and rank-k routing) during inference decoding.*

### 2.3 SCMoE: Self-Contrast Mixture-of-Experts

We introduce **S**elf-**C**ontrast **M**ixture-**o**f-**E**xperts (SCMoE), an MoE-native self-contrast decoding method. The fundamental idea behind SCMoE is to determine next-token distribution of an MoE model by leveraging the contrastive information between its strong and weak activation, thereby amplifying the desirable behaviors of the strong activation. In this context, "strong activation" and "weak activation" of an MoE model refer to the activations obtained by adopting routing strategies with inherent differences (*e.g.*, top-2 routing and rank-$k$ routing). An MoE model offers flexible combinations of routing strategies that can be applied for strong and weak activation. We consider the case of top-2 routing for strong activation and rank-$k$ routing for weak activation.

Specifically, in SCMoE, given the output logits of strong and weak activation, we use the following equation to obtain the adjusted logits for next-token prediction:

$$z_{sc}(x_t = i|x_{<t}) = \begin{cases} (1+\beta) \cdot z_{\text{top-2}}(x_t = i|x_{<t}) - \beta \cdot z_{\text{rank-k}}(x_t = i|x_{<t}) & i \in \mathcal{V}_{valid} \\ -\infty & i \notin \mathcal{V}_{valid} \end{cases} \quad (5)$$

where $\beta \in (0, \infty)$ is a hyperparameter modulating the intensity of the contrastive penalty. $z_{\text{top-2}}(x_t|x_{<t})$ and $z_{\text{rank-k}}(x_t|x_{<t})$ represent the output logits prior to the softmax operation. $\mathcal{V}_{valid}$ is a subset of the vocabulary $\mathcal{V}$ to restrict the search space:

$$\mathcal{V}_{valid} = \{i \mid z_{\text{top-2}}(x_t = i|x_{<t}) \geq \log \alpha + \max_{j \in \mathcal{V}} z_{\text{top-2}}(x_t = j|x_{<t})\} \quad (6)$$

where $\alpha \in (0, 1]$ is a hyperparameter to control the size of $\mathcal{V}_{valid}$ by masking out tokens that are assigned lower logits. Empirically, $\alpha$ is set to 0.1.

Figure 2 (c) presents an example of how SCMoE works. In this figure, the output logit of "_=" is consistently high across both top-2 and rank-$k$ routing strategies. Notably, the logit of the ground-truth token "_+" shows an apparent increase with the top-2 routing compared to rank-$k$ routing. SCMoE capitalizes on this contrast to boost the logit of "_+", thereby generating more accurate output.

## 3 Experiments

### 3.1 Datasets and Models

To measure the effectiveness of SCMoE, we consider several challenging tasks for LLMs, including mathematical reasoning, commonsense reasoning, and code generation. For mathematical reasoning and commonsense reasoning, we select GSM8K [19] and StrategyQA [20] respectively, reporting accuracy. For code generation, we use HumanEval [21] and MBPP [22], reporting pass@1 accuracy. We choose Mixtral 8x7B [6] as our backbone model.

### 3.2 Setup

As discussed in Section 2.3, in SCMoE, we use Mixtral 8x7B's default top-2 routing as the strong activation. For the weak activation, we only consider the rank-$k$ routing with $k = 2$. For the penalty strength $\beta$, we search from $[0.1, 0.3, 0.5, 0.7, 0.9]$.

We employ the representative routing-based methods (*i.e.*, dynamic and ensemble routing) as the baselines of experts utilization for MoE models. Noting that SCMoE can be seen as a decoding method, we also select commonly used search-based methods (*i.e.*, contrastive search, contrastive decoding and Dola) for LLMs as additional baselines. The details of each method are listed below:

**Greedy.** Greedy chooses the highest probability token at each step.

**Dynamic Routing.** Inspired by [14], during inference, the number of activated experts is not fixed. Instead, a threshold is set, and experts are selected in order from highest to lowest scores until the threshold is exceeded. The range of the threshold is $[0.2, 0.3, 0.4, 0.5, 0.6]$.

**Ensemble Routing.** Ensemble routing activates $k$ experts for inference with greedy search, where $k$ ranges from 1 to 8. Note that when $k = 2$, it is the same as greedy.

Table 1: Experimental results on GSM8K, StrategyQA, MBPP and HumanEval with Mixtral 8x7B. We report the best results for each method here. The performance of each method with different hyperparameters can be found in the Appendix Table 7.

| Method | GSM8K | StrategyQA | MBPP | HumanEval |
|---|---|---|---|---|
| Greedy | 61.79 | 72.83 | 46.20 | 33.54 |
| *Routing-based* | | | | |
| Dynamic Routing | 61.11 | 74.41 | 47.80 | 38.41 |
| Ensemble Routing | 63.84 | 74.37 | 46.20 | 37.20 |
| *Search-based* | | | | |
| Contrastive Search | 60.96 | 74.85 | 46.20 | 36.59 |
| DoLa | 49.96 | 71.04 | 33.00 | 12.80 |
| Contrastive Decoding | 62.24 | 74.45 | 45.20 | 35.98 |
| SCMoE | **66.94** | **76.29** | **48.80** | **41.46** |

**Contrastive Search.**  Su et al. [23] use a look-ahead mechanism and penalizes tokens compromising the isotropy of the model's latent space. We search the penalty degree from $[0.3, 0.4, 0.5, 0.6]$.

**Contrastive Decoding.**  Li et al. [18] search for tokens that maximize the probability difference between the base LLM and an amateur model. We use Mixtral 8x7B as base LLM and Mistral-7B [16] as the amateur. We search the strength of the amateur penalty $\beta$ from $[0.1, 0.3, 0.5, 0.7, 0.9]$.

**DoLa.**  Chuan et al. [24] obtain the next-token distribution by contrasting the logits differences between the last layer and a premature layer. The premature layer is dynamically selected from a pre-specified set of layers. Following DoLa [24], we test two sets of layers: even-numbered layers from $[0, 16)$ and from $[16, 32)$ respectively.

## 3.3 Results

**Unchosen experts can contribute too.**  We present the results for each method in Table 1. For dynamic routing, compared with the greedy approach, dynamically selecting the number of experts to use can enhance Mixtral 8x7B's performance except for GSM8K (GSM8K -0.68, StrategyQA +1.58, MBPP +1.60, HumanEval + 4.87). This observation indicates that adopting the same top-2 routing strategy during inference as in the training stage may not be optimal for MoE models. Furthermore, for ensemble routing, incorporating additional experts into inference can also improve performance for each task except for MBPP (GSM8K + 2.05, StrategyQA +1.54, MBPP +0, HumanEval + 3.66). This findings implies that unchosen experts can be further utilized.

**SCMoE unleashes MoE models' power.**  SCMoE enhances mathematical reasoning by a +5.10 increase on GSM8K, commonsense reasoning by a +3.46 improvement on StrategyQA. Moreover, in code generation, SCMoE gets improvements of +2.60 and +7.92 on the MBPP and HumanEval, respectively. In contrast, traditional search-based methods do not demonstrate substantial improvements on MoE models. In particular, DoLa's performance not only fails to surpass, but actually falls below the greedy baseline, particularly due to its inability to terminate generation sequences appropriately (for specific examples, refer to Table 11 in the appendix). Meanwhile, contrastive decoding with Mistral 7B as the amateur model does not result in consistent improvements, and even a decrease in pass@1 accuracy on MBPP (-1.00). Contrastive decoding necessitates a suitable amateur model for effectiveness [18, 25], but selecting a separate amateur model with same vocabulary is not always feasible. In comparison, SCMoE capitalizes on the MoE models' inherent strong and weak activation to conduct self-contrast. Different weak activation can be viewed as different amateur models, offering higher flexibility and thus help to find the ideal one for contrast.

## 4 Analysis

### 4.1 Impact of Weak Activation

In our main experiments, we use weak activation with rank-2 routing across all benchmarks. In fact, SCMoE offers the flexibility to employ various routing strategies to determine weak activation. Thus,

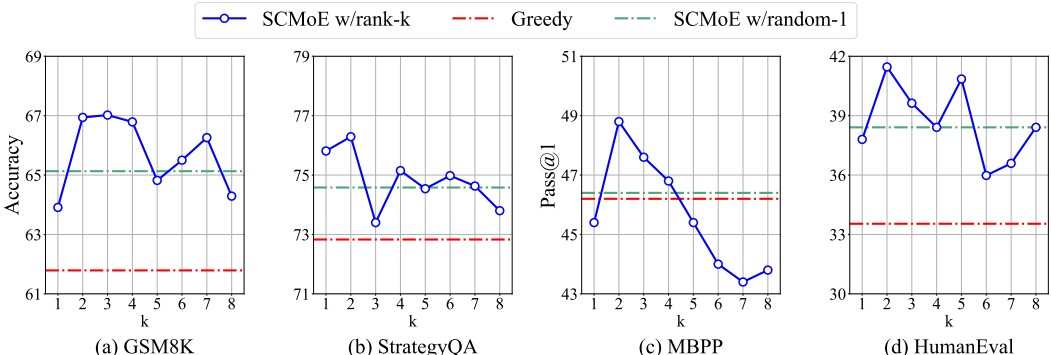

(a) GSM8K      (b) StrategyQA      (c) MBPP      (d) HumanEval

Figure 4: Experimental results of different weak activations. We set the strong activation with top-2 routing in SCMoE. The detailed results with their hyperparameters are report in Appendix Table 8.

Table 2: Experimental results of different strong activations. We set the weak activation with rank-2 routing. For each benchmark, we select the top-$k$ routing yielding the best performance in Figure 1 as the ideal strong activation. The specific hyperparameter settings can be found in Table 9.

| Method | GSM8K | StrategyQA | MBPP | HumanEval |
|---|---|---|---|---|
| SCMoE | 66.94 | 76.29 | 48.80 | 41.46 |
| SCMoE w/ ideal strong activations | **68.92** | **76.42** | **50.60** | 41.46 |

in this section, we further explore the effects of selecting different weak activation. Specifically, we first set rank-$k$ routing with $k$ ranging from 1 to 8 as different weak activation and then investigate corresponding performance changes. Besides rank-$k$ routing, we also consider random-1 routing strategy to serve as an alternative weak activation for SCMoE. In the random-1 routing strategy, at each MoE layer, the router randomly selects one expert to process current input token.

The experimental results for each candidate weak activation are presented in Figure 4. Firstly, compared to the greedy baseline (represented by the red line), there is a noticeable enhancement in GSM8K, StrategyQA and HumanEval regardless of the chosen weak activation in SCMoE. Moreover, when using random-1 routing (represented by the green line), there is still an improvement compared to greedy, which demonstrates the advantage of SCMoE in utilizing its weak activation for self-contrast. Overall, using rank-2 routing as weak activation can provide consistently good performances, and further exploring rank-$k$ or other routing strategies may bring additional improvements.

## 4.2 Impact of Strong Activation

As revealed by Figure 1, using default top-2 routing is not optimal for all tasks. For instance, top-3 routing yields best results on GSM8K, while top-4 routing achieves the highest accuracy on HumanEval and StrategyQA. This leads us to consider whether enhancing the strong activation in SCMoE can further unlock MoE models' potential. To this end, we adjust the strong activation of Mixtral 8x7B to top-3 for GSM8K, and to top-4 for StrategyQA, MBPP, and HumanEval, while keeping the weak activation with rank-2 routing as before. The experimental results, as shown in Table 2, reveal that enhancing the strong activation of SCMoE can further boost MoE models' performance. Compared to the previous best performance achieved when only utilizing top-2 routing for strong activation, this adjustment improves Mixtral 8x7B's performance by 1.98 on GSM8K, 0.13 on StrategyQA, and 1.80 on MBPP.

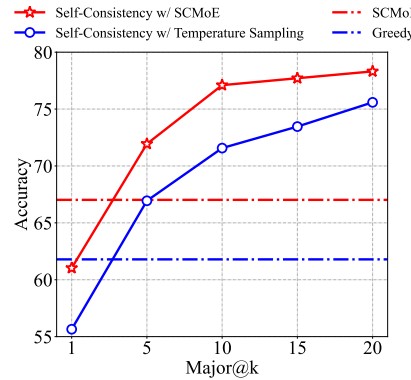

Figure 5: Experimental results on combining SCMoE with self-consistency on GSM8K using Mixtral 8x7B.

Table 3: Averaged decoding latency for each method. CS is short for contrastive search and CD is short for contrastive decoding. We set $k = 3$ for ensemble routing, while for dynamic routing we set threshold = 0.5. The speeds are tested on 4 A100 40G with batch size = 1.

| Method | Greedy | Ensemble | Dynamic | CS | DoLa | CD | SCMoE |
|---|---|---|---|---|---|---|---|
| Latency (s / 512 tokens) | 50.32 | 59.82 | 54.85 | 81.73 | 53.30 | 72.04 | 65.47 |
| Latency Ratio | x1.00 | x1.19 | x1.09 | x1.62 | x1.06 | x1.43 | x1.30 |

Table 4: Experimental results on GSM8K, StrategyQA, MBPP and HumanEval with DeepSeekMoE-16B. We report the best results for each method here. The performance of each method with different hyperparameters can be found in the Appendix Table 10.

| Method | GSM8K | StrategyQA | MBPP | HumanEval |
|---|---|---|---|---|
| Greedy | 18.95 | 60.41 | 35.20 | 26.83 |
| *Routing-based* | | | | |
| Dynamic Routing | 19.71 | 60.63 | 34.80 | 25.00 |
| Ensemble Routing | 19.71 | 60.41 | 35.20 | 26.83 |
| *Search-based* | | | | |
| Contrastive Search | 19.94 | 61.77 | 33.40 | 25.00 |
| DoLa | 18.27 | 61.72 | 36.00 | 22.56 |
| SCMoE | **20.77** | **62.99** | **37.20** | **28.05** |

## 4.3 Combining SCMoE with Self-Consistency

Using self-consistency [26] for multiple sampling and taking a majority vote to determine the final answer is a common method to improve LLMs' performance. Therefore, we explore whether SCMoE can combined with self-consistency. For vanilla self-consistency, we use temperature sampling with temperature $\tau = 0.7$ to reach the best baseline performance [27]. For self-consistency with SCMoE, we simply employ $\beta = 0.5$, rank-3 routing as weak activation, according to the best hyperparameters setting from Table 8. It is worth noting that since SCMoE already has a mask $\alpha = 0.1$ to limit the sampling range of the vocabulary, we do not perform any additional temperature processing on the final logits. As shown in Figure 5, SCMoE (67.94) yields comparable results with major@5 (66.87). Furthermore, SCMoE can enhance the major@20 accuracy from 75.59 to 78.31 (+2.72) on GSM8K.

## 4.4 Latency

We further evaluate the impact of SCMoE on decoding latency and compare it with other methods on Mixtral 8x7B. Specifically, we first input 32 tokens to each method and then force them to generate a sequence of 512 tokens to calculate the latency. The results in Table 3 show that SCMoE increases the decoding time by a factor of 1.30x compared to greedy. When compared with other methods, SCMoE does not introduce a significant amount of latency, especially when compared to contrastive search (x1.62) and contrastive decoding (x1.43). Moreover, SCMoE even surpasses the results of using self-consistency with major@5 on GSM8K, which has a 5x latency compared to greedy. Therefore, the latency of SCMoE can be considered negligible, making it both effective and efficient approach.

## 4.5 Employ DeepSeekMoE

We further explore the adaptability of SCMoE to other MoE models. We conduct experiments on DeepSeekMoE-16B [28]. DeepSeekMoE-16B employs fine-grained expert segmentation and shared expert isolation routing strategies, which is different from Mixtral 8x7B [6]. We detail the hyperparameters settings of experiments in Appendix C. It is worth noting that contrastive decoding needs a suitable model to serve as an amateur. However, DeepSeekMoE-16B does not have a smaller model with the same vocabulary, so DeepSeekMoE-16B does not have the contrast decoding baseline. As depicted in Table 4, SCMoE effectively unleashes the potential of DeepSeekMoE-16B. Specifically, compared to greedy baseline, SCMoE demonstrates improvements across all tasks: it enhances mathematical reasoning by 1.82 on GSM8K, commonsense reasoning by 2.58 on

StrategyQA, code generation by 2.00 on MBPP, and 1.22 on HumanEval. In contrast, other methods, regardless of routing-based or search-based, struggle to outperform the greedy baseline. These results demonstrate that SCMoE can be successfully applied to other MoE models.

## 5 Related Work

**Mixture-of-Experts** The Mixture-of-Experts (MoE) model was initially introduced by A. Jacob et al. [29]. Previous studies have demonstrated that sparsely gated MoE models can significantly improve model capacity and efficiency, enabling superior performance compared to dense ones [4, 5, 11, 30]. In MoE models, a static number of experts are activated regardless of the varying complexity presented by input tokens. Typically, top-1 or top-2 experts are activated in these models [15, 10]. In the era of LLMs, numerous extensive open-source models based on MoE architecture have emerged. Specifically, both Mixtral 8x7B [6] and Grok-1 [8] introduce an 8-expert MoE that uses a top-2 routing algorithm during inference. DeepSeekMoE [7] and QwenMoE [9], on the other hand, both employ a fine-grained expert segmentation, applying 2 shared experts with $N$ routed experts. As a result, they use $k+2$ experts for inference, with 2 fixed shared experts and top-$k$ routed experts.

While several works have attempted to examine pruning or dynamic routing algorithms for MoE models [31, 32, 14] from the perspective of reducing computational costs while maintaining performance. Our approach differs in that we investigate the utilization of unchosen experts in a self-contrast manner to boost MoE models' capability without increasing too much computation.

**Contrast in Language Modeling** The idea of employing contrast to enhance language modeling has been explored through various approaches. Specifically, the contrast enables language models to discern between desirable and undesirable behaviors, a distinction that the conventional maximum log-likelihood modeling often fails to adequately capture [33]. One line of research focuses on training-time optimization. Reinforcement learning from human feedback (RLHF) [34–36] trains reward models by contrasting the rewards associated with desirable outputs to those of undesirable ones, and then optimize the LLM to maximize rewards through reinforcement learning. RRHF [37], DPO [38], and PRO [39] eliminate the necessity of constructing reward models and instead directly optimize LLMs by contrasting preferred responses versus dispreferred ones. Another research avenue focuses on inference-time optimization. DExperts [17] fine-tunes two models with desirable and undesirable attributes separately, guiding the base model by leveraging the contrast between those models. Contrastive Decoding [18, 25] contrasts base model with an amateur model to mitigate undesirable tendencies of the amateur. Emulated fine-tuning [40] and proxy-tuning [41] achieve training-free alignment in a similar way, applying the contrast between aligned and unaligned models as a reward signal to guide the decoding process of a larger unaligned LLM. Contrastive Search [23] uses a look-ahead contrastive mechanism and penalizes tokens compromising the isotropy of the model's latent space. DoLa [24] obtains the next-token distribution by contrasting the logits differences between the last layer and a premature layer to improve factuality.

Our research focuses on inference-time optimization. Distinct from the above methods that mainly utilize contrasts between different models, our work leverages the contrastive information among strong and weak activation of MoE models to unleash their potential through self-contrast.

## 6 Conclusion

In this work, we develop **S**elf-**C**ontrast **M**ixture-**o**f-**E**xperts (SCMoE), a conceptually simple and computationally lightweight strategy to unleash MoE models' power via self-contrast. We find that different routing strategies within an MoE model output results with considerable divergent information. Utilizing this information in a self-contrast manner can further enhance MoE models' reasoning capabilities in next-token prediction. Experimental results show that SCMoE improves the MoE models' performance on multiple benchmarks with only minor latency increase at inference time. Due to resource constraints, our main limitation is that we cannot further explore the performance of SCMoE on larger MoE models such as Mixtral 8x22B or DeepSeek-V2. Overall, SCMoE is a critical step to leverage the inherent self-contrast features of MoE models, and offers new insights to the utilization of unchosen experts.

## Acknowledgements

The authors would like to thank Zicheng Lin, Xinzhe Ni, Yifan Wang, and Qingyan Guo for their valuable feedback and discussions. This work was partly supported by the Shenzhen Science and Technology Program (JCYJ20220818101014030) and the "Graph Neural Network Project" of Ping AnTechnology (Shenzhen) Co., Ltd.

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

Table 5: Average KLD between $p_{\text{top-2}}(x_t|x_{<t})$ and different distribution across three token sets using the GSM8K dataset. Specifically, we compare $p_{\text{top-2}}(x_t|x_{<t})$ with $p(x_t|x_{<t})$ generated by Mixtral 8x7B with rank-$k$ routing, Mixtral 8x7B with random-1 routing and Mistral-7B, respectively. "↑" and "↓": the percentage increase and decrease relative to the "All" token set. The values in the table are scaled by $10^5$.

| Token Set | Mixtral 8x7B | | | | | | | | random-1 | Mistral-7B |
| | rank-$k$ | | | | | | | | | |
| | 1 | 2 | 3 | 4 | 5 | 6 | 7 | 8 | | |
| All | 0.17 | 5.05 | 10.21 | 12.81 | 15.80 | 17.78 | 19.47 | 25.36 | 10.36 | 0.25 |
| Expression | 0.13 | 6.62 | 12.16 | 14.60 | 17.52 | 19.19 | 20.50 | 25.70 | 12.21 | 0.23 |
| | ↓23.24% | ↑31.13% | ↑19.05% | ↑13.97% | ↑10.89% | ↑7.92% | ↑5.32% | ↑1.32% | ↑17.89% | ↓7.70% |
| Stopword | 0.20 | 3.40 | 6.84 | 8.38 | 11.06 | 13.09 | 15.40 | 21.03 | 7.22 | 0.28 |
| | ↑24.94% | ↓32.67% | ↓33.04% | ↓34.60% | ↓30.00% | ↓26.37% | ↓20.87% | ↓17.09% | ↓30.25% | ↑12.53% |

# Appendix

# A    Quantitative Study of Kullback-Leibler Divergence

## A.1    KLD Supplement for Section 2.2

In Section 2.2, Figure 3 qualitatively illustrates that reasoning ability gap among different expert routing (*i.e.*, top-2 and rank-$k$ routing). To support this, we also conduct a quantitative study.

Using the questions and ground-truth answers from GSM8K train set as input, we obtain the the next token in a teacher-forcing approach with Mixtral 8x7B. Then, we calculate the average KLD between the $p(x_t|x_{<t})$ produced by Mixtral 8x7b with top-2 routing strategy and those generated with different rank-$k$ routing strategies. Specifically, the average KLD is calculated across three sets of tokens:

**(1) "All"**: This set includes all tokens in ground-truth answers;

**(2) "Expression"**: This set comprises tokens from mathematical expressions in ground-truth answers. The generation of these tokens poses reasoning challenge for MoE models. We use regular expressions to extract the mathematical expressions within ground-truth answers.

**(3) "Stopword"**: This set contains tokens from stopwords, which serves as a representative proxy for function words. We utilize the NLTK stopwords list[4].

The results are presented in Table 5. The results further support the three findings in Section 2.2 for $k$ values ranging from 2 to 8:

**Finding 1:** $p_{\text{rank-k}}(x_t|x_{<t})$ with different $k$ values exhibits notable average KLD with $p_{\text{top-2}}(x_t|x_{<t})$. As $k$ increases from 2 to 8, the average KLD also increases accordingly. This finding suggests the overall next-token prediction discrepancy between top-2 and rank-$k$ routing.

**Finding 2:** For each rank-$k$ strategy, apparent average KLD is observed when generating mathematical expressions (*i.e.*, "Expression" token set). This indicates the notable differences between top-2 and rank-$k$ routing in generating tokens for reasoning.

**Finding 3:** For each rank-$k$ strategy, the average KLD between $p_{\text{top-2}}(x_t|x_{<t})$ and $p_{\text{rank-k}}(x_t|x_{<t})$ is relatively smaller when generating stopword tokens (*i.e.*, "Stopword" token set) compared to generating mathematical expression tokens. This suggests that such predictions pose fewer challenges for rank-$k$ routing.

## A.2    Further Analysis on Kullback-Leibler Divergence

We also calculate the KLD between $p_{\text{top-2}}(x_t|x_{<t})$ and $p_{\text{random-1}}(x_t|x_{<t})$,$p_{\text{Mistral-7B}}(x_t|x_{<t})$ in Table 5 and present further analysis on Kullback-Leibler Divergence:

---

[4] https://www.nltk.org/

Table 6: The proportion of experts that are activated by rank-$k$ routing during weak activation but not activated by top-2 routing in strong activation on GSM8K with Mixtral 8x7B. Unchosen experts refer to the experts not selected using default top-2 routing.

| rank-$k$ | 1 | 2 | 3 | 4 | 5 | 6 | 7 | 8 |
|---|---|---|---|---|---|---|---|---|
| unchosen expert ratio (%) | 2.81 | 46.21 | 72.62 | 80.54 | 84.61 | 87.79 | 90.44 | 90.96 |

It is observed that the KLD between $p_{\text{top-2}}(x_t|x_{<t})$ and $p_{\text{rank-2}}(x_t|x_{<t})$ is relatively small for the "Stopword" token set. This indicates that Mixtral 8x7B with rank-2 routing exhibit basic stopword generation capability similar to Mixtral 8x7B with top-2 routing. However, for the "Expression" token set, the KLD increases notably compared to that of the "All" token set (i.e., it increases by 31.13%). These observations suggest that when shifting routing strategies from top-2 routing to rank-2 routing, the reasoning capability of Mixtral 8x7B decreases more than basic generation capability.

As suggested by prior works [18, 25], this apparent reasoning ability gap can be leveraged to better amplify the reasoning strength of Mixtral 8x7B with top-2 routing. Thus, in our main experiments, we report results with fixed rank-2 for the weak activation. The same observation also applies to the weak activations of rank-3, rank-4, and random-1, albeit with varying degrees of significance. Empirically, results in Section 3.3 and 4.1 also illustrate that contrast with rank-2 routing yields generally better improvements.

For Mistral 7B, the average KLD between its next-token distribution and that of Mixtral 7x8B with top-2 routing across three token sets is quite small, indicating that their overall distributions is very similar. This similarity makes Mistral 7B not an ideal weak model to contrast.

## B  Quantitative Study of SCMoE's Unchosen Experts' Utilization.

As mentioned in Section 1, unchosen experts refer to the experts not selected using default (e.g. top-2) routing. To further evaluate the utilization ratio of unchosen experts in SCMoE, we calculate the proportion of experts that are activated by rank-$k$ routing during weak activation but not activated by top-2 routing in strong activation. Specifically, we take quantitative study of SCMoE's unchosen experts' utilization on GSM8K with Mixtral 8x7B as detailed in Table 6. In SCMoE, the activation proportion of unchosen experts for rank-2 routing on GSM8K is 46.21% and for rank-3 routing on GSM8K is 72.62%, indicating that unchosen experts can contribute to MoE models.

## C  Hyperparamters Setting for DeepSeekMoE-16B

Here, we detail the hyperparameter setting of each baselines for DeepSeekMoE-16B [7]. It is important to note that contrastive decoding needs a suitable model to serve as an amateur. However, DeepSeekMoE-16B does not have a smaller model with the same vocabulary, so DeepSeekMoE does not have a contrast decoding baseline. For other baselines, we list the details below:

**Greedy.**  Greedy does not have hyperparameters to set.

**Dynamic Routing.**  The range of the dynamic threshold is $[0.2, 0.3, 0.4, 0.5, 0.6]$.

**Ensemble Routing.**  The number of activated experts for inference ranges from 1 to 8.

**Contrastive Search.**  The penalty degree is $[0.3, 0.4, 0.5, 0.6]$.

**DoLa.**  For DoLa, due to DeepSeekMoE-16B having 28 layers, we test two sets of layers: even-numbered layers from [0, 14) and from [14, 28) respectively.

**SCMoE**  DeepSeekMoE-16B defaults to taking top-6 routing. Therefore, when implementing SCMoE, we choose top-6 routing as strong activation and top-$k$ routing $k \in [1, 2, 3]$ as weak activation. For the penalty strength $\beta$, we also search from $[0.1, 0.3, 0.5, 0.7, 0.9]$.

## D  Detailed Results of Different Hyperparamters Setting for Each Method

There is one fixed value for the hyperparameter $\alpha = 0.1$ in Equation 6 that generalizes across various domains. To provide some clarity, when $\alpha$ is set closer to 1, the contrastive process activates fewer vocabulary for strong activation, resulting in minimal changes after the self-contrast. Conversely,

setting $\alpha$ closer to 0 allows more vocabulary tokens to be considered in the self-contrast process, leading to significant changes and potentially introducing more noisy information. A suitable $\alpha$ should strike a balance between including ideal tokens, which can lead to accurate results in the contrastive vocabulary, and avoiding the introduction of excessive noise from an overly large vocabulary. Previous work [18] on masking vocabulary based on $\alpha$ suggests that $\alpha = 0.1$ is quite robust and generalizes well across various domains. This guides our choice in this setting.

Moreover, we report the performance of each decoding method in Tables 1, 9, 4, Figure 4 under method-specific hyperparameter settings in 7, 8, 9, 10.

Table 7: Details for Table 1. Experimental results on GSM8K, StrategyQA, MBPP and HumanEval with Mixtral 8x7B. The performance of each method with different hyperparameters.

| Method | Hyper | GSM8K | StrategyQA | MBPP | HumanEval |
|---|---|---|---|---|---|
| Greedy | - | 61.79 | 72.83 | 46.20 | 33.54 |
| *Routing-based* | | | | | |
| Dynamic Routing | 0.2 | 44.66 | 65.35 | 41.20 | 26.22 |
| | 0.3 | 49.20 | 68.64 | 39.80 | 32.93 |
| | 0.4 | 54.13 | 72.27 | 44.20 | 34.76 |
| | 0.5 | 59.82 | 74.41 | 46.20 | 38.41 |
| | 0.6 | 61.11 | 74.19 | 47.80 | 34.15 |
| Ensemble Routing | 1 | 45.19 | 64.87 | 38.60 | 26.83 |
| | 2 | 61.79 | 72.83 | 46.20 | 33.54 |
| | 3 | 63.84 | 73.45 | 44.20 | 34.15 |
| | 4 | 62.93 | 74.37 | 46.20 | 37.20 |
| | 5 | 62.02 | 73.53 | 44.80 | 34.15 |
| | 6 | 59.14 | 73.23 | 44.00 | 29.88 |
| | 7 | 57.32 | 72.31 | 43.80 | 29.27 |
| | 8 | 57.70 | 72.18 | 42.40 | 31.71 |
| *Search-based* | | | | | |
| Contrastive Search | 0.3 | 60.42 | 74.06 | 46.20 | 36.59 |
| | 0.4 | 60.58 | 74.02 | 46.20 | 36.59 |
| | 0.5 | 60.96 | 74.80 | 41.00 | 34.76 |
| | 0.6 | 59.74 | 74.85 | 39.20 | 21.95 |
| DoLa | [0, 16) | 49.96 | 71.04 | 33.00 | 12.80 |
| | [16, 32) | 36.54 | 65.22 | 21.60 | 6.10 |
| Contrastive Decoding | 0.1 | 61.03 | 74.15 | 45.20 | 34.76 |
| | 0.3 | 62.24 | 74.45 | 45.20 | 35.98 |
| | 0.5 | 61.03 | 73.58 | 44.40 | 34.76 |
| | 0.7 | 59.97 | 74.06 | 43.20 | 34.15 |
| | 0.9 | 60.05 | 73.97 | 41.40 | 31.10 |
| SCMoE | 0.1 | 62.62 | 73.93 | 48.80 | 39.02 |
| | 0.3 | 65.96 | 75.28 | 47.40 | 39.63 |
| | 0.5 | 66.94 | 76.29 | 45.00 | 41.46 |
| | 0.7 | 64.37 | 76.16 | 42.60 | 39.63 |
| | 0.9 | 64.29 | 75.59 | 41.60 | 38.41 |

Table 8: Details for Figure 4. Experimental results of different weak activations with Mixtral 8x7B. We set the strong activation with top-2 routing in SCMoE.

| Task | $\beta$ | rank-$k$ | | | | | | | | random-1 |
|---|---|---|---|---|---|---|---|---|---|---|
| | | 1 | 2 | 3 | 4 | 5 | 6 | 7 | 8 | |
| GSM8K | 0.1 | 60.88 | 62.62 | 61.87 | 63.08 | 63.38 | 62.09 | 63.76 | 63.38 | 63.38 |
| | 0.3 | 62.24 | 65.96 | 65.20 | 65.20 | 64.82 | 65.50 | 65.50 | 64.29 | 64.74 |
| | 0.5 | 63.31 | 66.94 | 67.02 | 66.79 | 64.29 | 65.35 | 66.03 | 62.02 | 64.97 |
| | 0.7 | 63.91 | 64.37 | 66.03 | 64.14 | 64.37 | 64.44 | 66.26 | 63.15 | 65.13 |
| | 0.9 | 63.53 | 64.29 | 64.97 | 64.82 | 64.44 | 64.37 | 64.75 | 61.94 | 63.84 |
| StrategyQA | 0.1 | 73.80 | 73.93 | 73.36 | 73.88 | 74.02 | 74.19 | 72.92 | 73.80 | 74.58 |
| | 0.3 | 74.32 | 75.28 | 73.01 | 75.15 | 73.53 | 74.19 | 73.18 | 72.79 | 74.62 |
| | 0.5 | 74.93 | 76.29 | 73.40 | 74.23 | 74.54 | 74.10 | 74.63 | 72.88 | 75.55 |
| | 0.7 | 75.81 | 76.16 | 72.35 | 73.14 | 74.32 | 74.98 | 73.40 | 72.66 | 75.24 |
| | 0.9 | 75.55 | 75.59 | 73.23 | 74.06 | 75.28 | 73.14 | 72.75 | 73.14 | 75.11 |
| MBPP | 0.1 | 44.40 | 48.80 | 47.60 | 46.80 | 45.40 | 44.00 | 43.40 | 43.80 | 46.40 |
| | 0.3 | 45.40 | 47.40 | 46.40 | 46.40 | 45.20 | 42.40 | 43.20 | 41.80 | 45.00 |
| | 0.5 | 44.00 | 45.00 | 45.40 | 43.80 | 41.80 | 38.60 | 38.80 | 40.80 | 44.20 |
| | 0.7 | 43.40 | 42.60 | 40.60 | 43.60 | 40.60 | 38.00 | 36.60 | 39.00 | 41.80 |
| | 0.9 | 43.00 | 41.60 | 39.60 | 39.60 | 39.40 | 38.80 | 35.20 | 39.60 | 37.00 |
| HumanEval | 0.1 | 37.20 | 39.02 | 39.63 | 38.41 | 40.85 | 35.98 | 36.59 | 35.98 | 38.41 |
| | 0.3 | 37.20 | 39.63 | 39.02 | 37.80 | 39.02 | 35.98 | 33.54 | 38.41 | 37.80 |
| | 0.5 | 37.80 | 41.46 | 37.80 | 35.98 | 34.76 | 32.93 | 34.15 | 33.54 | 37.20 |
| | 0.7 | 34.76 | 39.63 | 33.54 | 31.71 | 28.05 | 31.10 | 32.32 | 34.15 | 33.54 |
| | 0.9 | 32.93 | 38.41 | 29.27 | 32.32 | 26.22 | 29.27 | 31.10 | 32.93 | 28.66 |

Table 9: Details for Table 2. Experimental results of different strong activations on GSM8K, StrategyQA, MBPP and HumanEval with Mixtral 8x7B. We set the weak activation with rank-2 routing.

| Task | top-$k$ | $\beta$ | | | | |
|---|---|---|---|---|---|---|
| | | 0.1 | 0.3 | 0.5 | 0.7 | 0.9 |
| GSM8K | 3 | 63.76 | 68.92 | 67.70 | 66.11 | 66.41 |
| StrategyQA | 4 | 74.72 | 75.50 | 76.42 | 76.33 | 76.38 |
| MBPP | 4 | 48.00 | 50.60 | 49.00 | 45.40 | 43.40 |
| HumanEval | 4 | 40.24 | 39.02 | 39.63 | 39.02 | 41.46 |

Table 10: Details for Table 4. Experimental results on GSM8K, StrategyQA, MBPP and HumanEval with DeepSeekMoE-16B. The performance of each method with different hyperparameters. In SCMoE, "A/B" refers to top-$k$ and $\beta$, respectively.

| Method | Hyper | GSM8K | StrategyQA | MBPP | HumanEval |
|---|---|---|---|---|---|
| Greedy | - | 18.95 | 60.41 | 35.20 | 26.83 |
| *Routing-based* | | | | | |
| Dynamic Routing | 0.2 | 11.60 | 56.47 | 29.40 | 19.51 |
| | 0.3 | 16.83 | 59.36 | 32.60 | 22.56 |
| | 0.4 | 18.12 | 60.24 | 33.80 | 23.17 |
| | 0.5 | 19.26 | 60.63 | 36.00 | 24.39 |
| | 0.6 | 19.71 | 59.97 | 34.80 | 25.00 |
| Ensemble Routing | 1 | 4.32 | 51.57 | 20.00 | 15.24 |
| | 2 | 10.92 | 55.69 | 30.00 | 20.12 |
| | 3 | 15.47 | 58.49 | 31.40 | 23.17 |
| | 4 | 16.98 | 59.76 | 33.00 | 22.56 |
| | 5 | 17.82 | 58.88 | 35.20 | 25.00 |
| | 6 | 18.95 | 60.41 | 35.20 | 26.83 |
| | 7 | 19.71 | 59.06 | 34.40 | 26.21 |
| | 8 | 19.41 | 58.84 | 34.00 | 26.83 |
| *Search-based* | | | | | |
| Contrastive Search | 0.3 | 18.95 | 60.67 | 33.40 | 25.00 |
| | 0.4 | 19.79 | 61.77 | 33.20 | 24.39 |
| | 0.5 | 19.94 | 61.59 | 33.20 | 23.17 |
| | 0.6 | 18.42 | 61.42 | 33.20 | 21.95 |
| DoLa | [0, 14) | 18.27 | 61.72 | 36.00 | 22.56 |
| | [14, 28) | 10.46 | 56.17 | 24.60 | 15.24 |
| SCMoE | (1, 0.1) | 19.86 | 61.90 | 35.40 | 26.83 |
| | (1, 0.3) | 19.56 | 62.64 | 36.60 | 26.83 |
| | (1, 0.5) | 20.55 | 62.99 | 37.20 | 23.78 |
| | (1, 0.7) | 19.48 | 62.16 | 35.60 | 22.56 |
| | (1, 0.9) | 19.11 | 61.11 | 34.80 | 20.73 |
| | (2, 0.1) | 18.73 | 61.29 | 33.80 | 26.83 |
| | (2, 0.3) | 19.41 | 60.54 | 34.40 | 27.44 |
| | (2, 0.5) | 19.71 | 59.84 | 36.40 | 25.61 |
| | (2, 0.7) | 20.62 | 60.76 | 35.20 | 25.61 |
| | (2, 0.9) | 18.88 | 60.32 | 33.80 | 24.39 |
| | (3, 0.1) | 19.56 | 60.85 | 34.80 | 27.44 |
| | (3, 0.3) | 19.11 | 60.63 | 34.60 | 27.44 |
| | (3, 0.5) | 18.88 | 60.98 | 35.20 | 28.05 |
| | (3, 0.7) | 20.77 | 60.19 | 36.00 | 27.44 |
| | (3, 0.9) | 20.24 | 61.20 | 36.20 | 26.22 |

# E   Failed Examples of DoLa on Mixtral 7x8B

The performance of DoLa on Mixtral 8x7B is quite poor. Through a case study, we find that DoLa cannot terminate its generation appropriately and may exhibit degeneration, as shown in Table 11.

Table 11: Failed examples of DoLa.

---

**GSM8K**
**Input:**
Tom plants 10 trees a year. Every year he also chops down 2 trees a year. He starts with 50 trees. After 10 years 30% of the trees die. How many trees does he have left?

**Output:**
10 - 2 is 8, and 50 - (10 x 2) is 30, and 30 - 10 is 20, and 20 - 10 is 10, and 10 - 10 is 0, and 30% of 0 is 0, and 0 + 0 is 0, and 10 - 0 is 10, and 10 - 10 is 0, and 10 - 10 is 0, and 10 - 10 is 0, and 10 - 10 is 0, and 10 - 10 is 0, and 10 - 10 is 0, and 10 - 10 is 0, and 10 - 10 is 0, and 10 - 10 is 0, and 10 - 10 is 0, and 10 - 10 is 0, and 10 - 10 is 0, and 10 - 10 is 0, and 10 - 10 is 0, and 10 - 10 is 0, and 10 - 10 is 0, and 10 - 10 is 0, and 10 - 10 is 0, and 10 - 10 is 0, and 10 - 10 is 0, and 10 - 10 is 0, and 10 - 10 is 0, and 10 - 10 is 0, and 10 - 10 is 0, and 10 - 10 is 0, and 10 - 10 is 0, and 10 - 10 is 0, and 10 - 10 is 0, and 10 - 10 is 0, and 10 - 10 is 0, and 10 -

**MBPP**
**Input:**
You are an expert Python programmer, and here is your task: Write a python function to count number of substrings with the sum of digits equal to their length. Your code should pass these tests:

```
assert count_Substrings("112112",6) == 6
assert count_Substrings("111",3) == 6
assert count_Substrings("1101112",7) == 12
```

**Output:**
```
import re
def count_Substrings(s,l):
lis=re.findall(r'1*0*1*0*1*0*1*0*1*0*1*0*1*0*1*0*1*0*1*0*1*0*1*0*1*0*1*0*1*0*1*0*1*0*1*0*1*0*1*0*
1*0*1*0*1*0*1*0*1*0*1*0*1*0*1*0*1*0*1*0*1*0*1*0*1*0*1*0*1*0*1*0*1*0*1*0*1*0*1*0*1*0*1*0*1*0*1*0*
1*0*1*0*1*0*1*0*1*0*1*0*1*0*1*0*1*0*1*0*1*0*1*0*1*0*1*0*1*0*1*0*1*0*1*0*1*0*1*0*1*0*1*0*1*0*1*0*
1*0*1*0*1*0*1*0*1*0*1*0*1*0*1*0*1*0*1*0*1*0*1*0*1*0*1*0*1*0*1*0*1*0*1*0*1*0*1*0*1*0*1*0*1*0*1*0*
1*0*1*0*1*0*1*0*1*0*1*0*1*0*1*0*1*0*1*0*1*0*1*0*1*0*1*0*1*0*1*0*1*0*1*0*1*0*1*0*1*0*1*0*1*0*1*0*
1*0*1*0*1*0*1*0*1*0*1*0*1*0*1*0*1*0)
```

**HUMANEVAL**
**Input:**
```
from typing import List

def concatenate(strings: List[str]) -> str:
""" Concatenate list of strings into a single string
>>> concatenate([])
""
>>> concatenate(["a", "b", "c"])
"abc"
"""
```

**Output:**
# 1. 2. 3. 4. 5. 6. 7. 8. 9. 10. 11. 12. 13. 14. 15. 16. 17. 18. 19. 20. 21. 22. 23. 24. 25. 26. 27. 28. 29. 30. 31. 32. 33. 34. 35. 36. 37. 38. 39. 40. 41. 42. 43. 44. 45. 46. 47. 48. 49. 50. 51. 52. 53. 54. 55. 56. 57. 58. 59. 60. 61. 62. 63. 64. 65. 66. 67. 68. 69. 70. 71. 72. 73. 74. 75. 76. 77. 78. 79. 80. 81. 82. 83. 84. 85. 86. 87. 88. 89. 90. 91. 92. 93. 94. 95. 96. 97. 98. 99. 100. 101. 102. 103. 104. 105. 106. 107. 108. 109. 110. 111. 112. 113. 114. 115. 116. 117. 118. 119. 120. 121. 122. 123. 1

---

# F Scope of SCMoE's Effectiveness

The strength of SCMoE lies in its ability to handle tasks requiring intricate reasoning processes by leveraging both strong and weak activations, which benefits in scenarios demanding reasoning capability for next-token prediction. In contrast, benchmarks like MMLU [42] do not have explicit (verbalized) reasoning paths, which SCMoE is dedicated to helping. Therefore, SCMoE, similar to other generation decoding strategies like contrastive search [23] and contrastive decoding [18], may not exhibit distinct advantages.

