# OpenReview forum: "Unchosen Experts Can Contribute Too: Unleashing MoE Models’ Power by Self-Contrast"
_NeurIPS.cc/2024/Conference — NeurIPS 2024 poster_

### Official Review · Reviewer_Vqkj · 2024-06-15

**Soundness:** 2
**Presentation:** 3
**Contribution:** 3
**Rating:** 7
**Confidence:** 4

**Summary:**

This paper proposes a training-free strategy that utilizes unchosen experts in a self-contrast manner during inference. It can be seen as a decoding method utilizing divergent information from different routing strategies. This method introduces slightly more latency overhead and improves the performance of various tasks through experimental evaluation.

**Strengths:**

* The paper is well put together with clear insights and well-articulated motivation.
* The problem studied in the paper is well-motivated, it is known that expert selection in MoE is not trivial.
* The idea is novel, and the illustration of the methodology is easy to follow.

**Weaknesses:**

* This inference method employs two models, one with top-2 routing and the other with rank-k routing. It produces double memory overhead which is significant, especially for larger models.
* The choosing of α in Equation 6. seems non-trivial, it would be great to provide some insight on this.

**Questions:**

* The extra latency cost is minor, is it because the two-model inference is parallelly implemented?
* The “+2=5 (...)” in Figure 2.(c) is a little bit confusing to me, could you explain further about it?

**Limitations:**

Yes.

---

> ### Author Rebuttal · Authors · 2024-08-06
>
> Thank you for your valuable feedback. We appreciate that you highlight the motivation, idea and the clarity of our work. We provide detailed responses to address your specific concerns as outlined below.
>
> > W1: This inference method employs two models, one with top-2 routing and the other with rank-k routing. It produces double memory overhead which is significant, especially for larger models.
>
> R1:
> Thank you for your question. In fact, SCMoE incurs only minimum memory overhead because it does not use two models. Instead, it performs both weak activation forward passes and strong activation forward passes simultaneously using a single model (As we explained in Figure 2 and Section 2.3).
> Specifically, this additional memory is due to SCMoE doubling the size of the KV cache caused by additional rank-k routing. This additional memory scales linearly with the sequence length. For instance, for a sequence length of 2048 using BF16, the additional memory amounts to approximately 256.0MB. Given the model size of nearly 86GB, this represents a marginal increase of about 0.3%. This minor increment underscores the feasibility of our approach in practical deployments.
>
> > W2: The choosing of $\alpha$ in Equation 6. seems non-trivial, it would be great to provide some insight on this.
>
> R2:
> Thank you for your insightful feedback. There is one fixed value for the hyperparameter $\alpha$=0.1 in Equation 6 that generalizes across various domains. To provide some clarity, when $\alpha$ is set closer to 1, the contrastive process activates fewer vocabulary for strong activation, resulting in minimal changes after the self-contrast. Conversely, setting  $\alpha$ closer to 0 allows more vocabulary tokens to be considered in the self-contrast process, leading to significant changes and potentially introducing more noisy information.
>
> A suitable $\alpha$ should strike a balance between including ideal tokens, which can lead to accurate results in the contrastive vocabulary, and avoiding the introduction of excessive noise from an overly large vocabulary. Previous work [1] on masking vocabulary based on $\alpha$ suggests that "$\alpha$ = 0.1 is quite robust and generalizes well across various domains."  This guides our choice in this setting.
>
> We will include these details in the next version. We appreciate your thoughtful feedback and hope this response addresses your concern.
>
> [1] Contrastive decoding: Open-ended text generation as optimization.
>
>
> > Q1: The extra latency cost is minor, is it because the two-model inference is parallelly implemented?
>
> R3:
> Thank you for your valuable question. In fact, we do not use two separate models for inference, we only use a single MoE model's strong activation and weak activation for self-contrast.
>
> Specifically, upon receiving a test input, we duplicate it. As a result, we have two identical test inputs. During the forward pass of the MoE model, we apply top-2 routing strategy to one test input and rank-$k$ routing strategy to the other input. This approach allows us to achieve both strong and weak activation within a single forward pass, thereby reducing the extra latency cost.
>
> > Q2: The “+2=5 (...)” in Figure 2.(c) is a little bit confusing to me, could you explain further about it?
>
> R4: Thank you for your valuable question. Figure 2(c) presents an example of how SCMoE operates. The complete question and corresponding answer for this example are depicted in Figure 3. In Figure 2(c), the model is tasked with predicting the next token (represented by the "?" symbol with the green background indicating the unknown next token). Utilizing the SCMoE method, the model successfully predicts the next token to be "+" (represented by "+" with the green background).
> The notation "2 = 5 (...)" following the "+" represents part of the answer that has not yet been generated (the complete answer can be found in Figure 3). In fact, it is unnecessary to include "2 = 5 (...)" in Figure 2(c). We appreciate your suggestion and will clarify this in the next version.
>
> Thank you for your insightful reviews! We sincerely appreciate your kind words regarding the novelty of our work.

---

> > ### Comment · Reviewer_Vqkj · 2024-08-10
> >
> > Thank the authors for their answers to my concerns. They have adequately answered every question I raised to my satisfaction, and therefore, I will keep my current score of an acceptance.

---

### Official Review · Reviewer_cUsK · 2024-06-25

**Soundness:** 4
**Presentation:** 4
**Contribution:** 4
**Rating:** 8
**Confidence:** 4

**Summary:**

This paper proposes a novel approach called Self-Contrast Mixture-of-Experts (SCMoE) to improve the utilization and performance of Mixture-of-Experts (MoE) models. The key contributions are:

1. Exploratory studies showing that increasing the number of activated experts in MoE models does not always improve output quality and that different routing strategies lead to substantially different output distributions.

2. The SCMoE method, which leverages unchosen experts in a self-contrast manner during inference. It determines next-token probabilities by contrasting outputs from strong and weak activation using the same MoE model.

3. Experimental results demonstrating that SCMoE consistently enhances the reasoning capabilities of Mixtral 8x7B across various domains.

4. Combining SCMoE with self-consistency yields further gains.

5. The proposed SCMoE method is conceptually simple, computationally lightweight, and incurs minimal latency compared to greedy decoding.

**Strengths:**

1. The paper is well-motivated through exploratory studies, showing that simply increasing the number of activated experts in MoE LLMs can harm performance and that expert models tend to inhibit rather than strengthen each other.

2. The proposed SCMoE method is simple and intuitive, using the difference between the logits of stronger and weaker models. This approach incurs minimal performance overhead and is straightforward to implement.

3. Despite its simplicity, SCMoE consistently yields performance gains on evaluation benchmarks, showcasing its effectiveness.

4. The selected set of evaluation tasks, while not extensive, includes representative tasks from the most important domains of LLM applications, making the experimental results meaningful and easy to interpret. The study also examines multiple mainstream MoE LLMs, such as Mixtral and DeepSeekMoE.

5. The paper is well-written and well-presented.

**Weaknesses:**

1. The idea of SCMoE is somewhat similar to the cited paper "Contrastive decoding: Open-ended text generation as optimization (Li et. al. [20])," although I understand that SCMoE is better adapted to MoE LLMs and has lots of novelties in other aspects. The authors could provide more discussion in the related work section about the similarities and differences between the two approaches.

2. The paper lacks clarity on how certain hyperparameters are chosen. For example, in Section 3.2, the authors state that "for the weak activation, we only consider the rank-k routing with k=2" but do not provide an explanation for why k cannot be 3, 4, or 5. It would be helpful to know if choosing k=2 is motivated by faster inference times, as the top 2 models will be used regardless.

**Questions:**

1. See Weakness 2.

2. I am curious why Mixtral has decreased performance when averaging over more expert models. Could the authors give some explanations? It is because of something specific to pretraining methodology of Mixtral/DeepSeekMoE models?

---

> ### Author Rebuttal · Authors · 2024-08-06
>
> Thank you for your recognition of the novelty, simplicity and effectiveness of our work. This is a great honor for us. We aim to address your concerns below.
>
> > W1: The idea of SCMoE is somewhat similar to the cited paper "Contrastive decoding: Open-ended text generation as optimization (Li et. al. [20])," although I understand that SCMoE is better adapted to MoE LLMs and has lots of novelties in other aspects. The authors could provide more discussion in the related work section about the similarities and differences between the two approaches.
>
> R1:
> Thank you for your insightful feedback. The motivation behind SCMoE is to investigate the utilization of unchosen experts. Our final approach leverages the unchosen experts by contrasting predictions of strong activation with weak activation. We acknowledge that there are some existing works sharing the similar spirit in using a contrastive objective function. We have already discussed them in the Related Work section ("Contrast Language Modeling", Lines 279-299). To further address your concern, we give more detailed explanations regarding the similarities and differences between our work and [1] as follows:
>
> *Similarity:*
> - Both SCMoE and [1] belong to the category of inference-time optimization with contrast.
> - Both approaches contrast two different distributions to obtain a better distribution to enhance model generation performance.
>
> *Difference:*
> - To obtain two different distributions, [1] requires two different models—an expert model and an amateur model—making the selection of a suitable expert model and amateur model combination a challenge, as it involves ensuring vocabulary consistency, and various expert-amateur model parameter scale combinations.
> - In contrast, SCMoE takes advantage of the architecture characteristic of MoE models by employing different routing strategies to directly obtain two different distributions. Therefore, SCMoE essentially utilizes only one single model, supports dynamic distribution combinations, and effectively employs the unchosen experts in MoE models.
>
> [1] Contrastive decoding: Open-ended text generation as optimization.
>
> > W2 & Q1: The paper lacks clarity on how certain hyperparameters are chosen. For example, in Section 3.2, the authors state that "for the weak activation, we only consider the rank-k routing with k=2" but do not provide an explanation for why k cannot be 3, 4, or 5. It would be helpful to know if choosing k=2 is motivated by faster inference times, as the top 2 models will be used regardless.
>
> R2:
> Thank you for your insightful feedback. To clarify, the choice of $k$ in the rank-$k$ routing strategy is guided by the criterion of maintaining consistency in generating general tokens, such as stopwords, while providing a notable contrast for tokens requiring reasoning capabilities.
>
> In Appendix A (Lines 429-473), we qualitatively illustrate the average KLD between  $p_{\text{top-2}}(x_{t}|x_{<t})$ and different distributions $p_{\text{rank-k}}(x_{t}|x_{<t})$. It is observed that the KLD between $p_{\text{top-2}}(x_{t}|x_{<t})$ and $p_{\text{rank-2}}(x_{t}|x_{<t})$ is relatively small for the "Stopword" token set. This indicates that Mixtral with rank-2 routing exhibits basic stopword generation capability similar to Mixtral with top-2 routing. However, for the "Expression" token set, the KLD increases notably compared to that of the "All" token set (i.e., it increases by 31.13%). These observations suggest that when shifting routing strategies from top-2 routing to rank-2 routing, the reasoning capability of Mixtral decreases more than basic generation capability.
>
> As suggested by prior works [1], this apparent reasoning ability gap can be leveraged to better amplify the reasoning strength of Mixtral with top-2 routing. The same observation also applies to the weak activations of rank-3, rank-4, and random-1, albeit with varying degrees of significance.
>
> Empirically, results in Sections 3.3 and 4.1 illustrate that rank-2 routing generally yields better improvements. Hence, rank-2 routing serves as a practical and versatile configuration applicable across various domains. To consistently validate the effectiveness of SCMoE, we present our findings using a fixed rank-2 for weak activation.
> While rank-2 is effective, it may not be optimal for all tasks. Future work could explore how to adaptively set the size of weak activation rank-k for different user queries.
>
> [1] Contrastive decoding: Open-ended text generation as optimization.
>
> > Q2: I am curious why Mixtral has decreased performance when averaging over more expert models. Could the authors give some explanations? It is because of something specific to pretraining methodology of Mixtral/DeepSeekMoE models?
>
> R3:
> Your question is very insightful. We believe this is still an open question, and we explain our understanding as follows. Similar to your hypothesis, we also believe it is due to something specific to the pretraining methodology. In addition to the standard Language Modeling loss, additional losses such as Expert-Level Balance Loss and Device-Level Balance Loss are used to address the MoE models' loading balance issue, ensuring stable training. This load balancing loss in pretraining may enable different experts in MoE models to excel at different domains / tasks but not all. When "non-specialized" experts are activated, their contributions may become noise in the final ensemble, diluting the strengths of truly specialized experts. Based on this observation, we consider utilizing these "non-specialized" (wrt. unchosen ) experts in a self-contrast manner in order to benefit in scenarios demanding reasoning capability for next-token prediction (Lines 111-126).
>
> Thank you for your insightful suggestion! We sincerely appreciate your praise for the novelty of our work.

---

### Official Review · Reviewer_bydP · 2024-07-21

**Soundness:** 3
**Presentation:** 4
**Contribution:** 3
**Rating:** 6
**Confidence:** 3

**Summary:**

This paper introduces SCMoE, a decoding time algorithm which can be applied off the shelf to boost MoE models' performance by contrasting the chosen experts in strong and weak activations. Experiment results show that this algorithm has an empirical advantage over baselines methods in coding, commonsense knowledge, and math benchmarks.

**Strengths:**

Originality:
This paper proposes an original method in MoE decoding. The method is straightforward to understand and easy to deploy.

Quality:
This paper is of high quality. The paper is well written with good demonstrations and clear math. The proposed method shows better performance across all presented benchmarks compared to baseline methods.

Clarity:
As discussed in the last point, this paper is very clear in idea demonstration and methodology illustration. Experiment results are also well organized.

Significance:
This paper is significant in contributing to the after-training development of MoE models.

**Weaknesses:**

1.	Commonsense reasoning in StrategyQA, which is a multi-hop reasoning dataset about world knowledge, does not seem to be so strongly related to coding and math which requires formal reasoning compared to formal logic (see [1], which mainly focuses on logic, algorithm, and math as reasoning tasks). Do you anticipate performance boost in logic reasoning tasks?

2.	Relatedly to the last question, do you think the proposed algorithm will also help general world knowledge reasoning without implicit complex reasoning chains such as MMLU?


3.	Is it possible to find the ideal strong activation in real-life workflows only given the user query?

[1] Zhu, K., Chen, J., Wang, J., Gong, N. Z., Yang, D., & Xie, X. (2023). Dyval: Dynamic evaluation of large language models for reasoning tasks. In The Twelfth International Conference on Learning Representations.

**Questions:**

See above

**Limitations:**

The authors briefly mentioned limitations of their paper in the conclusion section but they could further discuss other limitations including limitation of datasets evaluated etc.

---

> ### Author Rebuttal · Authors · 2024-08-06
>
> Thank you for your positive feedback! We appreciate your praise of the Originality, Quality, Clarity and Significance of our work, which is a great encouragement for us. We would like to address your concerns below:
>
> > W1: Commonsense reasoning in StrategyQA, which is a multi-hop reasoning dataset about world knowledge, does not seem to be so strongly related to coding and math which requires formal reasoning compared to formal logic (see [1], which mainly focuses on logic, algorithm, and math as reasoning tasks). Do you anticipate performance boost in logic reasoning tasks?
>
> R1:
> We conduct experiments on three logical reasoning tasks as mentioned in [1]: abductive logic, boolean logic, and deductive logic. Since Mixtral achieves nearly 100% accuracy in boolean logic, we only present the results for abductive logic and deductive logic below.
>
> |  Dataset  | Greedy | Dynamic | Ensemble |  CS  | DoLa |  CD  | SCMoE |
> |:---------:|:------:|:-------:|:--------:|:----:|:----:|:----:|:-----:|
> | Abductive |  68.2  |   77.0  |   70.2   | 69.2 | 74.0 | 81.2 |  87.6 |
> | Deductive |  78.4  |   80.8  |   78.4   | 77.4 | 84.6 | 84.6 |  86.4 |
>
> From these results, it is evident that SCMoE provides significant performance improvements in abductive and deductive logic. This finding verifies that SCMoE can also bring performance boost in logical reasoning tasks.
>
> [1] Dyval: Dynamic evaluation of large language models for reasoning tasks.
>
> > W2: Relatedly to the last question, do you think the proposed algorithm will also help general world knowledge reasoning without implicit complex reasoning chains such as MMLU?
>
> R2:
> The strength of SCMoE lies in its ability to handle tasks requiring intricate reasoning processes by leveraging both strong and weak activations, which benefits in scenarios demanding reasoning capability for next-token prediction (Lines 111-124). In contrast, benchmarks like MMLU do not have explicit (verbalized) reasoning paths, which SCMoE is dedicated to helping. Therefore, SCMoE, similar to other generation decoding strategies like Contrastive Search (CS) and Contrastive Decoding (CD), may not exhibit distinct advantages. We will further discuss and clarify the applicability of SCMoE in the next version.
>
> > W3: Is it possible to find the ideal strong activation in real-life workflows only given the user query?
>
> R3:
> The topic of finding the ideal strong activation in real-life workflows based solely on the user query is indeed a topic worthy of further exploration in future research.
> Previous work [2] offers some insights regarding this problem. It suggests that "Harder tasks need more experts" and proposes a routing algorithm that can dynamically adjust the number of experts activated based on the difficulty of the problem.
> While for our method SCMoE, it can achieve better performance than the ideal strong activation as shown in Figure 1, reducing the reliance on searching for ideal strong activation. The performance of SCMoE can be further boosted when we can obtain the ideal strong activation as shown in Table 2, this indicates that our method can be further enhanced with other ideal activation searching method, and we leave that for future research.
>
> [2] Harder tasks need more experts: Dynamic routing in MoE models.
>
> Thanks again for your valuable feedback!

---

> > ### Comment · Reviewer_bydP · 2024-08-11
> >
> > Thank you for addressing my concerns! I’ll keep my positive score for the paper.

---

### Official Review · Reviewer_KVjj · 2024-07-21

**Soundness:** 2
**Presentation:** 3
**Contribution:** 2
**Rating:** 6
**Confidence:** 4

**Summary:**

The paper explores to leverage the contrastive information existing between different routing strategies of the MoE model to facilitate a better token decoding during inference in a training-free fashion. The paper is built upon two interesting observations:
(1) increasing the number of activated experts does not necessarily improve and can even degrade the output quality (which increases and then drops might be due to noise/irrelevancy of other experts).
(2) output distributions from an MoE model using different routing strategies substantially differ (which is obvious due to pre-training setting with a routing strategy).

More specifically, for ScMoE the next-token probabilities are determined by contrasting the outputs from strong and weak activation using the same MoE mode. The authors conducted experiments using Mixtral on GSM8K, StrategyQA, MBPP and HumanEval and illustrate some noticeable performance benefits on GSM8K.

**Strengths:**

The paper brings several interesting strength to community:

1. The U-shape performance and top-k routing policy is indeed interesting and changing k wrt. dataset for best performance can encourage MoE designs with adaptive inference based on task difficulty.
2. The authors proposal to estimate next-token probabilities by contrasting the outputs from strong and weak activation using the same MoE model is interesting and I am surprised to see the impressive performance gain (>5 points) on GSM8K.
3. The authors presents some findings (ln. 111-120 and Appendix) as well as a comprehensive ablation experiments which make the paper well-grounded and comprehensive.
4. The author additionally include some latency related issues which is the major bottleneck of their proposed method.

**Weaknesses:**

While the paper is well-written and comprehensive, I have following comments/questions/concerns related to the paper:

1. One question- the author submitted numbers for Mixtral default routing strategy seems to be far from the numbers reported by the Mistral authors (Table 2 from https://arxiv.org/pdf/2401.04088). What could be the possible reason for that (curiosity and not a negative point)?
2. The author must perform the memory bottleneck for storing additional activations for their purpose of using two multiple routing strategy in SCMoE. I believe the memory bottleneck (as well as latency) of decoding will be substantially high while dealing with long-context tasks.
3. I was wondering if the contrastive behavior can be enforced explicitly by using a small finetuning data and a contrastive loss? What are the authors thought about this?
4. What is the consistency of the findings KLD for reasoning tasks across some other math reasoning tasks apart from GSM8K?
5. Another major weakness of the work is the tuning of the hyperparameter `\beta` for every dataset? Can authors find some discover a single `\beta` which  should work well enough for say a task category (math reasoning/commonsense).

I still find the work interesting and willing to increase the score with fair discussion during rebuttal.

**Questions:**

See above.

**Limitations:**

See above.

---

> ### Author Rebuttal · Authors · 2024-08-06
>
> Thank you for your positive feedback and willingness to take further discussions with us. We appreciate your praise of the insights (S1, S2) and the detailed experimental setup (S3, S4) in our work. We provide point-to-point response to address your concerns as follows:
>
> > W1: One question- the author submitted numbers for Mixtral default routing strategy seems to be far from the numbers reported by the Mistral authors (Table 2 from [1]). What could be the possible reason for that (curiosity and not a negative point)?
>
> R1:
> We would like to clarify that the performance reported in Table 2 of the Mixtral [1] are using the self-consistency with major@8 (as detailed in Section 3 of their experimental setup). For direct results without using self-consistency, you can refer to Table 3 in Mixtral [1]. We hope this addresses your concern.
>
> [1] Mixtral of Experts
>
>
> > W2: The author must perform the memory bottleneck for storing additional activations for their purpose of using two multiple routing strategy in SCMoE. I believe the memory bottleneck (as well as latency) of decoding will be substantially high while dealing with long-context tasks.
>
> R2:
> In practice, the memory overhead and decoding latency introduced by SCMoE are within acceptable bounds. The primary cause of the additional memory usage is the simultaneous employment of both strong activation (e.g. , top-2 routing) and weak activation (e.g. , rank-2 routing) in an MoE model. Specifically, this additional memory is required for storing additional KV cache, which scales linearly with the sequence length. For a sequence length of 2048, the additional memory amounts to approximately 256.0MB using BF16. Given the model size of nearly 86GB, this represents a marginal increase of about 0.3%. As for decoding latency, it consistently remains at x1.30, and this ratio does not increase with longer lengths. This minor increment underscores the feasibility of our approach in practical deployments.
>
> > W3: I was wondering if the contrastive behavior can be enforced explicitly by using a small finetuning data and a contrastive loss? What are the authors thought about this?
>
> R3:
> This is a very interesting point. Enforcing contrastive behavior explicitly by using a small finetuning data and a contrastive loss is indeed a promising direction of future work. This is a straightforward approach but would greatly increase training cost. We consider that maybe applying it at the late stage of training using a small amount of fine-tuning data is definitely worth trying.
>
> Nevertheless, the key insight we want to share is that we can more effectively utilize the MoE's unchosen experts in a simple yet effective training-free approach since they are already trained and loaded into memory. In terms of further more effective utilization of these experts combined with additional training methods, due to limited space, we leave for future work..
>
> > W4: What is the consistency of the findings KLD for reasoning tasks across some other math reasoning tasks apart from GSM8K?
>
> R4:  Based on your suggestion, we conduct experiments on a more challenging math reasoning dataset MATH [3]. Due to computational resource limitations, we randomly sample 500 examples from the MATH test set for our experiments. We perform a quantitative study of KLD on the MATH dataset using the same settings as in Appendix A. The results are shown in the table below:
>
> |  Token Set |  rank-k |         |         |         |         |         |        |        |
> |:----------:|:-------:|:-------:|:-------:|:-------:|:-------:|:-------:|:------:|:------:|
> |            |    1    |    2    |    3    |    4    |    5    |    6    |    7   |    8   |
> |     All    |   0.31  |   7.81  |  10.43  |  16.13  |  18.84  |  21.96  |  23.84 |  30.92 |
> | Expression |   0.26  |   9.27  |  11.68  |  17.17  |  19.83  |  22.84  |  25.20 |  31.73 |
> |            | -16.69% | +18.77% | +11.99% |  +6.47% |  +5.25% |  +3.99% | +5.74% | +2.60% |
> |  Stopword  |   0.38  |   5.82  |   8.42  |  13.11  |  15.98  |  19.54  |  21.49 |  29.96 |
> |            | +22.50% | -25.42% | -19.28% | -18.70% | -15.19% | -11.03% | -9.85% | -3.10% |
>
> The results are consistent with our observations on the GSM8K dataset (Lines 446-455), reaffirming that MoE models employing top-2 and rank-k routing strategies exhibit distinct generation behaviors. This finding further supports the analytical foundation of SCMoE in utilizing contrastive information effectively.
> We also compare the results of SCMoE and other methods on the MATH dataset. The experimental results are shown in the table below. The results indicate that SCMoE continues to be effective even on more challenging math reasoning tasks.
>
> |   Dataset  | Greedy | Dynamic | Ensemble |  CS  | DoLa |  CD  | SCMoE |
> |:----------:|:------:|:-------:|:--------:|:----:|:----:|:----:|:-----:|
> | Math (500) |  20.2  |   21.0  |   21.2   | 21.4 | 16.4 | 20.6 |  22.4 |
>
> [3] Measuring Mathematical Problem Solving With the MATH Dataset
>
> > W5: Another major weakness of the work is the tuning of the hyperparameter \beta for every dataset? Can authors find some discover a single \beta which should work well enough for say a task category (math reasoning/commonsense).
>
> R5:
> Thank you for your question. In fact, the performance of different $\beta$ can be seen in Table 7 of Appendix D, where $\beta$ = 0.5 already achieves the best results on GSM8K, StrategyQA, and HumanEval (except for MBPP). Therefore, $\beta$ = 0.5 may be sufficient for most scenarios and further tuning could achieve optimal performance. Meanwhile, as shown in Table 7, other baseline methods also face the same issue. However, for SCMoE, we believe automatically adjusting the $\beta$ based per each instance or token can be a direction worth exploring, which we leave for future work.
>
> We appreciate your positive feedback and willingness to take further discussions and increase the rating. Hope that our response can address your concerns.

---

> ### Author Response · Authors · 2024-08-11
> **A Gentle Reminder**
>
> Dear Reviewer KVjj,
>
> Thank you once again for your constructive comments on our submission. Your feedback is really valuable to the improvement of our paper. As the discussion phase nears its conclusion, we eagerly await any further comments or questions you might have.
>
> We hope that our responses have adequately addressed your concerns. If you find our revisions satisfactory, we would greatly appreciate it if you could consider raising the score of your assessment. If there are still issues to be addressed, please let us know, and we would be more than happy to engage in further discussion.

---

> ### Comment · Area_Chair_c9zc · 2024-08-12
>
> Dear reviewer,
>
> Can you let the author and us know if you've read the rebuttal, and if you have any further comments?
>
> Thanks,
>
> AC

---

> > ### Comment · Reviewer_KVjj · 2024-08-12
> > **Response to Authors.**
> >
> > Thank you for the thorough response, I encourage you to polish the submitted version. I will raise my score.

---

### Decision · Program_Chairs · 2024-09-25

**Decision:**

Accept (poster)

**Comment:**

This paper proposes a simple inference strategy that utilizes unchosen experts in a MoE framework. The paper starts with two interesting observations: (1) increasing the number of activated experts does not necessarily improve and can even degrade the output quality; (2) output distributions from an MoE model using different routing strategies substantially differ. These observations led them to propose the Self-Contrast Mixture-of-Experts (SCMoE), which modifies the decoding strategy by contrasting outputs from strong and weak activation using the same MoE model. This method improves the performance of various tasks (GSM8K, StrategyQA, MBPP and HumanEval) of Mixtral 8x7B with a slight latency overhead. Combining SCMoE with self-consistency yields further gains.

All reviewers agree that the paper presents very insightful observations, proposes a simple, off-the-shelf decoding strategy that's massively
effective. The findings and the method can benefit anyone studying and deploying MoE models immediately.